# CASP microdomain formation requires cross cell wall stabilization of domains and non-cell autonomous action of LOTR1

Andreas Kolbeck[1†], Peter Marhavý[1†‡], Damien De Bellis[1,2], Baohai Li[3§], Takehiro Kamiya[3], Toru Fujiwara[3], Lothar Kalmbach[1#], Niko Geldner[1*]

[1]Department of Plant Molecular Biology, University of Lausanne, Lausanne, Switzerland; [2]Electron Microscopy Facility, University of Lausanne, Lausanne, Switzerland; [3]Department of Applied Biological Chemistry, The University of Tokyo, Tokyo, Japan

*For correspondence: niko.geldner@unil.ch

[†]These authors contributed equally to this work

Present address: [‡]Umeå Plant Science Centre (UPSC), Department of Forest Genetics and Plant Physiology, Swedish University of Agricultural Sciences (SLU), Umeå, Sweden; [§]College of Environmental and Resource Sciences, Zhejiang University, Hangzhou, China; [#]Sainsbury Laboratory, University of Cambridge, Cambridge, United Kingdom

Competing interest: The authors declare that no competing interests exist.

**Abstract** Efficient uptake of nutrients in both animal and plant cells requires tissue-spanning diffusion barriers separating inner tissues from the outer lumen/soil. However, we poorly understand how such contiguous three-dimensional superstructures are formed in plants. Here, we show that correct establishment of the plant Casparian Strip (CS) network relies on local neighbor communication. We show that positioning of Casparian Strip membrane domains (CSDs) is tightly coordinated between neighbors in wild-type and that restriction of domain formation involves the putative extracellular protease LOTR1. Impaired domain restriction in *lotr1* leads to fully functional CSDs at ectopic positions, forming 'half strips'. LOTR1 action in the endodermis requires its expression in the stele. LOTR1 endodermal expression cannot complement, while cortex expression causes a dominant-negative phenotype. Our findings establish *LOTR1* as a crucial player in CSD positioning acting in a directional, non-cell-autonomous manner to restrict and coordinate CS positioning.

## Introduction

Plants mine the surrounding soil for water and dissolved minerals to sustain their growth and to complete their life cycle. Likened to inverted guts, roots evolved crucial epithelial functions — selective uptake and diffusion barriers — in order to generate and sustain body homeostasis in variable and harsh environments. In gut epithelial cells, the required separation of outside lumen and inside tissues is achieved through the formation of tight junctions. These specialized membrane domains form ring-like domains between the apical (gut lumen) and basal (blood stream) sides. Close adhesion of both adjacent plasma membranes is achieved by tight interaction of occludins, claudins, and cadherins, organized into tight and adherens junctions, forming impermeable and mechanically-resistant barriers. In plants, a functionally similar diffusion block is achieved by Casparian strips (CS), highly localized impregnations of the cell wall, the plant's extracellular matrix, in root endodermal cells.

Similar to tight/adherence junctions, a specialized membrane domain (Casparian Strip membrane Domain, CSD) forms a precise ring in transversal and anticlinal membranes of elongated endodermal cells. This domain acts as a molecular fence that separates the endodermal plasma membranes into peripheral (outer) and central (inner) domains, as highlighted by the polar distribution of nutrient transporters (*Alassimone et al., 2010*; *Bao et al., 2019*; *Winter et al., 2007*; *Takano et al., 2010*; *Ma et al., 2007*). CASPARIAN STRIP DOMAIN PROTEINs (CASPs) are specifically targeted to this domain and form a stable matrix for the subsequent lignification machinery (*Roppolo et al., 2011*; *Lee et al., 2013*; *Hosmani et al., 2013*; *Kalmbach et al., 2017*; *Barbosa et al., 2019*). Lignification of the primary cell wall of CSs extends through the entire apoplast including cross-linkage to the middle

lamella. This is illustrated in cell wall digestions leaving only the resistant lignified tissues of CSs and xylem visible as a fishnet-like matrix (*Enstone et al., 2003*).

The CS and tight/adherens junctions must form a contiguous network in order to fulfill their role as functional diffusion barriers. However, we still poorly understand how such a supracellular structure is coordinated within the endodermal cell layer. Although we are ignorant of the mechanism, the precisely opposing localization of CSDs evidences that endodermal cells coordinate the positioning of their membrane domains across the cell wall space. In epithelial cells, adherens junctions are first initiated by cell-to-cell contact mediated by nectins and cadherins and kept in place by firm attachment to the actin cytoskeleton (*Rajasekaran et al., 1996*; *Itoh and Bissell, 2003*). In plant cells, their cell wall matrices separate neighboring cell by at least 100–200 nm, which prevents such a direct interaction of membrane-bound proteins. Thus, a more complex positioning mechanism that requires local signals from directly adjacent neighbors could ensure coordination of CSDs between neighbors.

Many components necessary for correct CS formation have been identified in recent years, but few show impacts on CS positioning per se. Mutation of components in the recently discovered barrier surveillance mechanism causes interrupted, but correctly localized CSDs (*Pfister et al., 2014*; *Doblas et al., 2017*; *Nakayama et al., 2017*; *Alassimone et al., 2016*). Loss of the endodermal differentiation transcription factor MYB36 abolishes the formation of the entire CSD while disruption of CS cell wall localised ESB1 causes unstable, but still correctly localised domains (*Kamiya et al., 2015*; *Liberman et al., 2015*; *Hosmani et al., 2013*). The recently published *lotr2* mutant revealed the importance of a member of the exocyst complex, *EXO70A1*, in mediating targeted secretion of CASP proteins to the pre-established CSD (*Kalmbach et al., 2017*). However, the fragmented domains visible in this mutant are non-functional and unstable. Thus, based on current data, it is unclear whether formation of functional CSDs at ectopic positions is even possible.

Here, we report that the recently discovered *lotr1* mutant forms fully functional CSDs outside the endodermal-endodermal cell interface in. We establish that neighboring endodermal cells are absolutely required to form stable and continuous CSDs, and we highlight close coordination of CSD microdomains during the early stages of domain formation. In addition, we demonstrate that *LOTR1* is crucial for restriction of the CSD and that it acts independently of any known pathway for CSD formation. We show that this new type of putative cell wall protease is expressed in the stele, represents a novel, non-cell-autonomous and directional signaling component that controls CSD stability in the endodermis. Thus, we demonstrate that signaling, both within and between cell layers is crucial for coordinating CS positioning.

## Results

### Ectopic CASP deposits in Lotr1 are fully functional CSDs

During their initial characterization of the *lotr1* mutant, *Li et al., 2017* reported a novel 'patchy' phenotype, where large patches of the Casparian strip membrane domain marker CASP1::CASP1-GFP occurred outside of the regular median CS position. Further analysis revealed an apoplastic delay phenotype attributed to large gaps that could be observed in the CS upon lignin stain. We performed a more detailed analysis of the domain marker CASP1::CASP1-GFP in this mutant background to search for disruptions in the CSD that could explain these gaps in lignification. Confirming the earlier findings (*Li et al., 2017*), we found large patches of CASP1-GFP occurring in endodermal membranes that were never observed in wildtype plants (*Figure 1A–C*). Whereas previously reported to occur only facing pericycle cells, we found similar patches to also occur towards the outer cortex (*Figure 1A–C*, *Figure 1—figure supplement 1A*), although quantification revealed a clear preference for the endodermal-pericycle interface (*Figure 1—figure supplement 1B*). In addition to these patches, we found disruptions of the central CSD which occurred primarily at cell corners with nearby ectopic CASP deposits (*Figure 1B*), coherent with the reported gaps in CS lignin. The overall correctly positioned and largely continuous central CSD in *lotr1* suggests that the core-localization mechanism for the CSD is still functional. We speculate that the change in cell shape and cell walls at junctions might favor patch formation in *lotr1*. If patch formation starts simultaneously with initial focalization of CASPs into micro-domains (known as the string-of-pearls stage), nearby ectopic CASP deposits might inhibit the growth of the central domain leading to the observed large discontinuities. In order to better understand the nature and origin of these patches, we followed CASP deposition throughout

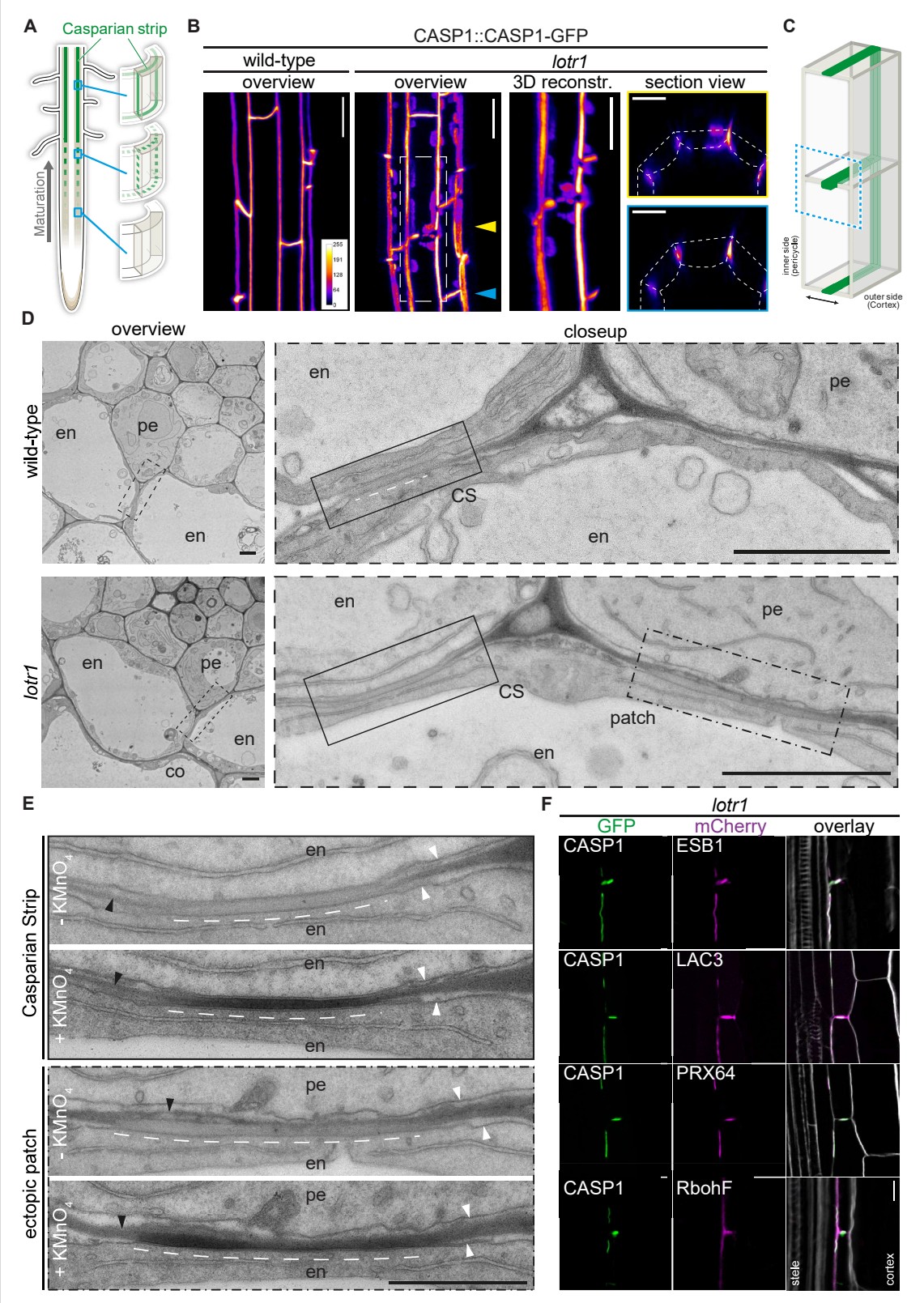

**Figure 1.** Ectopic domain formation in *lotr1*. (**A**) Schematic representation of endodermal maturation with stages of Casparian Strip (CS) development (green). Elongated endodermal cells focalize CASP1-GFP into aligned micro-domains which are subsequently fused into a continuous ring around each endodermal cell. (**B**) CASP1-GFP localization in wildtype and *lotr1* with focus on patch localization. Images depict maximum intensity projections of z-stack *Figure . and 3D* reconstruction image depicts region indicated in *lotr1* overview image, section view images are annotated by yellow (top)

*Figure 1 continued on next page*

*Figure 1 continued*

and blue (bottom) arrows. Overview images scale bars: 20 µm; 3D reconstruction & section views scale bars: 10 µm. (**C**) 3D schematic for visualization of median endodermal cell section with annotated cell faces. (**D**) TEM section of wildtype and *lotr1* obtained at 2 mm from root tip. Overview of section with focus on regular CS and adjacent ectopic patch. Please note the tight plasma membrane-attachment to the CS that is a further feature of CS presence. Closeup indicated by dashed black line in overview image; en = endodermis, pe = pericycle, co = cortex, scale bars = 2 µm. (**E**) Consecutive TEM sections of *lotr1* at 2 mm from root tip, with and without $KMnO_4$ staining, position indicated in lower panels of (**D**). Dark deposits indicate electron-dense $MnO_2$ precipitation caused by reaction with lignin. Note that in the case of ectopic patches, the staining is restricted to the endodermal side of the cell wall. In addition, plasma membrane-attachment to the CS is only occurring at the endodermal side. en = endodermis, pe = pericycle, white arrows indicate plasma membranes, black arrows highlight cell wall middle lamellae, interspaced white line depicts extend of CS-like cell wall morphology, scale bar = 1 µm. (**F**) Localization of enzymes necessary for CS lignification in wildtype Col-0 and *lotr1*, pictures depict cells as seen in (**C**). Cellulosic cell walls are stained with Calcofluor White and depicted in white in overlay images, scale bar = 10 µm.

The online version of this article includes the following figure supplement(s) for figure 1:

**Figure supplement 1.** Establishment of ectopic domain formations in *lotr1*.

endodermal differentiation. In both wildtype and *lotr1*, CASP1-GFP aggregated into aligned micro-domains in the cell's median (*Figure 1—figure supplement 1C*). During this initial focalization step, we observed occurrence of ectopic CASP islands in *lotr1*, indicating a role of LOTR1 during this early step of defining the CS location. Ectopic CASP deposits grew over time and were able to fuse into larger patches, similarly to the central micro-domains combining into a continuous ring. We did not observe degradation of these patches nor formation of new patches once the central domain was fused, confirming that these patches are stable once formed and their establishment is restricted to the short developmental window in which the cells determine their CS location. Consequently, LOTR1 likely acts in this early phase as an inhibitor of ectopic domain formation. We then investigated whether these patches represent fully functional domains, that is whether they are able to guide lignification enzymes to these sites. Deposition of lignin can be easily identified in electron microscopy, where CSs are visible as a clearly defined, homogenous stretch of cell wall spanning the entire apoplastic space (*Roppolo et al., 2011*; *Fujita et al., 2020*; *Figure 1D*). Ultra-structure analysis of *lotr1* revealed similar cell wall modifications in pericycle-facing endodermal membranes (*Figure 1D*). To our surprise, and in contrast to regular CSs, this ectopic lignin patch was restricted to the endodermal part of the cell wall, stopping precisely at the middle lamella, effectively creating half of a Casparian Strip. Such a strict spatial confinement of lignification on a nanometer scale is quite surprising since monolignols and ROS, both essential for lignin polymerization, are thought to be mobile in cell walls and lignin polymerization itself not to be under direct enzymatic control. To our knowledge, such a strictly restricted lignification encompassing only half of a thin primary has not been observed previously. Our finding therefore suggests a very local and directed lignin polymerization in the CS, where each endodermal cell is responsible for lignifying their part of the cell wall to build a functional CS barrier. A functional CSD is also characterized by a tight attachment of the underlying membrane, which is revealed in EM by the mild plasmolysis of the plasma membrane introduced through sample preparation, and which is absent at the CSD. Perfectly matching the cell autonomous nature of the ectopic patches, tight attachment of the endodermal plasma membrane, but not the pericycle plasma membrane was observed.

In order to further support our hypothesis that *lotr1* ectopic patches are fully functional domains, we compared consecutive electron micrograph sections with and without potassium permanganate ($KMnO_4$) staining to check for lignification. Depositing electron dense material by reaction with lignin sidechains, $KMnO_4$ highlights lignin presence as darkening of cell walls in Casparian strips and protoxylem vessels (*Stein et al., 1992*; *Yamashita et al., 2016*). Ectopic patches in *lotr1* displayed KMnO4 staining on the endodermal side of the half-strips, while the opposite pericycle cell walls were unstained (*Figure 1E*). This correlated with CS-typical cell wall appearance (more homogenous, less electron-dense, slightly thickened) observed in unstained consecutive cuts, confirming lignin presence at these ectopic domains. Notably, several samples displayed CS-type cell wall morphology with accompanying membrane attachment and lignification throughout the entire endodermal cell corner, stretching from the median endodermal-cell interface all the way to the endodermal-pericycle cell walls (*Figure 1—figure supplement 1D*). Although ectopic lignification is a hallmark of many Casparian Strip mutants, this was not described for *lotr1*. Furthermore, enhanced lignification itself does not result in firm membrane attachment outside the regular CS. Consequently, this points to a

drastically enlarged CSD indicative of an impaired ability to restrict CSD growth in *lotr1,* agreeing with a presumed role of LOTR1 as inhibitor of domain formation.

The regular CSD recruits a multitude of enzymes required for restricted monolignol polymerization to form the CS. The tightly restricted lignin accumulation at ectopic *lotr1* patches suggested similar recruitment of these enzymes. We therefore co-expressed the CSD marker CASP1-GFP together with typical members of this lignin machinery in *lotr1* to confirm their recruitment to ectopic CSDs. Endodermis specific PEROXIDASE 64 (PRX64) and LACCASE 3 (LAC3) displayed a clear co-labeling in both central and ectopic domains while NADPH oxidase RbohF - the main generator of ROS for monolignol polymerization - was also found to be present, although it did not specifically accumulate at the CSDs (*Figure 1F*). Finally, we analyzed the localization of dirigent-domain protein ESB1 in *lotr1*. Although it was previously reported to be absent from ectopic deposits (*Li et al., 2017*). We found ESB1 to accumulate at the sites of ectopic CASP1-deposition when expressed as ESB1-mCherry under its native promoter together with CASP1-GFP. Dirigent-domain proteins are thought to be a key component in restricting and directing lignin polymerization in the cell wall (*Hosmani et al., 2013*). Our results of ESB1 localization therefore agree with PER64 and LAC3 labeling, as well as the tight restriction of lignin at these patches, proving these ectopic formations to be fully functional, but ectopically formed CSDs.

## Focalization of CASPs is coordinated between cells

The fact that an ectopic CSD patch towards a non-endodermal neighbor can only induce formation of a half-strip suggested that two neighboring endodermal cells absolutely need to align their respective CSDs in order to lignify the entire CS cell wall space. To ensure this, a mechanism would need to be in place that promotes formation of a CSD at places where a neighboring cell is forming one and/ or inhibits CSD formation in the absence of a neighboring CSD. We therefore investigated the initial focalization of CASP proteins at high resolution. When properly aligned along the z-axis, we were able to identify two distinct membrane signals present between two neighboring endodermal cells (*Figure 2A and C*). Strikingly, at the string-of-pearls stage, CASP1-GFP intensity along the length of both signals already showed a strong positive correlation ($R^2$ = 0.535, p = 1.612 e-9) (*Figure 2B*), indicating coordination of CSD formation across the cell wall already at this stage. To validate our assumption that these paired signals represent the signals of neighboring cell membrane domains, we checked the position of CS cell wall protein ESB1 and cell wall tracer Propidium Iodide (*Figure 2C and D*). In contrast to CASP1-GFP, ESB1-mCherry localized in a single line in between the observed CASP-labeled membranes in both early and late CSDs (*Figure 2C*). To further confirm that these GFP signals truly represented separate membranes, we decided to employ cell-specific laser ablation to destroy one of the cells, arguing that this should remove only one half of the paired signal. Indeed, upon ablation, GFP signal was specifically lost in the ablated cell, whereas the adjacent signal of the intact neighboring cell was unchanged (*Figure 2D–F*, *Figure 2—figure supplement 1*). Together, our data demonstrate the observed double signals to be adjacent CSDs of neighboring endodermal cells, whose initial formation is tightly coordinated between cells. We then investigated whether currently known mutants such as *sgn1*, *sgn3,* and *esb1* would display defects in CASP microdomain coordination (*Figure 2G*). However, all these mutants showed wildtype like coordination between adjacent CSDs. Interestingly, *lotr1* also showed well-coordinated deposition of micro-domains at the Casparian strip position. Therefore, while LOTR1 appears to be necessary to restrict, unpaired, ectopic domain establishment, its loss-of function does not seem to affect the process of coordination at the correct central position where the future CS between neighboring cells is forming. This is corroborated by the lack of CS half-strips at the regular median position between endodermal cells in *lotr1* (*Figure 2G*).

## Endodermal cells require a neighboring cell to stabilize CASPs

The close correlation of CASP micro-domains in neighboring endodermal membranes could be explained by an unknown feedback control mechanism that stabilizes microdomains upon presence of neighboring domains and de-stabilizes them in their absence. Such a mechanism would immediately guide CSD formation towards endodermis-endodermis surfaces. In order to test this model, we ablated individual endodermal cells in the elongation zone, prior to their transition to differentiation. This enabled us to compare CSD establishment with and without a functional neighbor in the same cell. Initially, we observed CASP deposition in a string-of-pearls-typical pattern in membranes facing

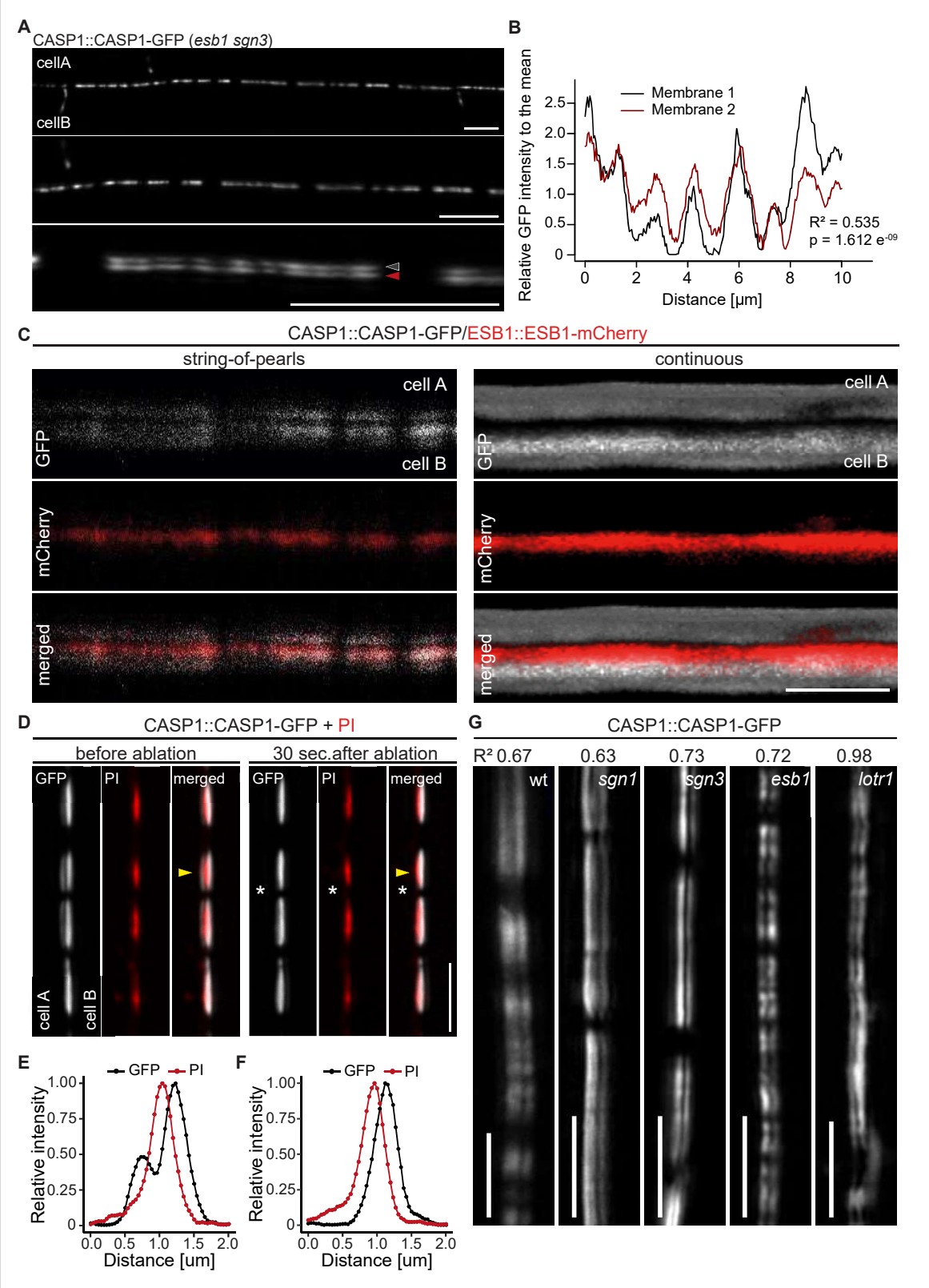

**Figure 2.** CASP deposition is coordinated between neighboring endodermal cells. (**A**) Double membrane phenotype visible with CASP1::CASP1-GFP during string-of-pearls stage of CS development, scale bars depict 10 µm. Red and gray arrows indicate separate membranes measured in (**B**). (**B**) Comparison of GFP intensity between adjacent membranes indicated in (**A**). Pixel intensity was measured along each membrane and relative GFP intensity was adjusted to mean intensity of each membrane. Correlation of original intensities (R-squared) was determined by fitting of a linear

*Figure 2 continued on next page*

*Figure 2 continued*

model with indicated probability of fit (p-value). (**C**) Localization of CSD marker CASP1-GFP in comparison to cell wall protein ESB1-mCherry at early and mature CSs, scale bar depicts 5 µm. (**D**) Double membrane phenotype of CASP1-GFP before and after ablation of one adjacent endodermal cell in early CSs. Cell walls (red) are stained with Propidium iodide (PI), yellow arrows indicate intensity profiles measured in F-I; asterisks indicate ablated cell, scale bars = 5 µM. (**E + F**) Quantification of CASP1-GFP and PI intensity before and after cell ablation. Intensity was quantified in a 7-pixel wide line indicated in (**D**) and (**E**). (**G**) Double membrane phenotype in CS mutants, scale bars = 5 um. $R^2$ value depicts Spearman rho correlation coefficients of fluorescence of 5-pixel wide lines following each membrane.

The online version of this article includes the following figure supplement(s) for figure 2:

**Figure supplement 1.** CASP1-GFP membrane coordination.

living and dead cells alike (*Figure 3A and D*). However, only membranes facing intact neighbors were able to form a continuous strip, whereas domains towards ablated cells did not progress past the string-of-pearls stage. When we ablated all adjacent cells, the single intact cell was still able to express and focalize CASPs but was unable to establish any continuous domains (*Figure 3C*). We also noticed a strong difference in signal intensity between membranes facing living and dead cells, where GFP signal increased rapidly in CSDs towards intact neighbors, but not ablated cells (*Figure 3B*). This discrepancy is likely a result of an inability to recruit additional CASP proteins to the initially formed micro-domains. This suggested a control point during CSD establishment where a neighboring signal is required to progress from the string-of-pearls stage and commit to formation of a CS. We had to exclude that isolating endodermal cells from their neighbors by ablation might cause stresses that influence overall endodermal differentiation and not just a destabilization of CASP recruitment to the membrane. We therefore used a CASP1 transcriptional reporter, driving a nuclear fluorescent protein. Upon ablation of all surrounding endodermal cells a similar CASP promoter activity compared to non-affected cells was observed, indicating that progress of differentiation per se was not halted in the isolated cell (*Figure 3—figure supplement 1A*). To rule out possible general effects on secretion or localization of plasma membrane proteins, we then ablated cells in double marker lines expressing CASP1-GFP and the generic SYP122-3xmCherry plasma membrane marker, both under the control of the CASP1 promoter. Again, this clearly established that isolated cells still express from the CASP1 promoter. Moreover, they accumulate SYP122 on all cell sides, in contrast to CASP1, which accumulates only at plasma membrane domains facing a living neighbor (*Figure 3D*).

We then thought to confirm our findings by means that do not involve destruction of neighboring cells. To do so we generated inducible complementation lines for MYB36, a key transcription factor of endodermal differentiation (*Liberman et al., 2015*; *Kamiya et al., 2015*). *myb36* loss-of-function mutants are unable to activate expression of key CSD proteins, for example CASPs, and consequently lack a CSD. Using very low level of inducer, we generated sporadic MYB36 activation to enable some, but not all cells to start differentiation and initiate CS formation. Expectedly, we observed a sporadic, patchy expression pattern of CSD domain marker CASP1-GFP upon induction, with some samples displaying expression in isolated, single cells, while others showed cell groups with a near complete CSD network (*Figure 3E*). Confirming our ablation results, we found that isolated cells were able to deposit CASP1-GFP in typical micro-domains but were incapable of forming a stable and continuous CSD, similar to what was observed using cell ablation (*Figure 3E*, left). If two adjacent cells started to deposit CASPs, we observed an increase in GFP signal only in the domain between these cells, followed by fusion of the initial micro-domains into a continuous strip (*Figure 3E*, right). The domains not facing a CASP1 expressing neighbor did not progress beyond a string-of-pearls stage and eventually lost the CASP1-GFP signal (*Figure 3F*, *Figure 3—figure supplement 1B*). We conclude that wildtype plants appear to require signals from neighbors to form stable microdomains and commit to a CS. *lotr1* is currently the only known mutant that can form functional microdomains in the absence of neighbors. We therefore asked whether *lotr1* might be able to fuse micro-domains at the side facing the dead neighbor into continuous strips. Yet, when we ablated cells in *lotr1*, we observed a similar disappearance of the central CSD facing the ablated cell (*Figure 3—figure supplement 1C*). One explanation could be that LOTR1 function is redundant for destabilization of unpaired domains in the correct, central position, but is required at surfaces facing non-endodermal neighbors. Indeed, we did observe formation of stable, *lotr1*-typical patches on the side facing its dead neighbor, although they occurred in membranes facing living pericycle cells.

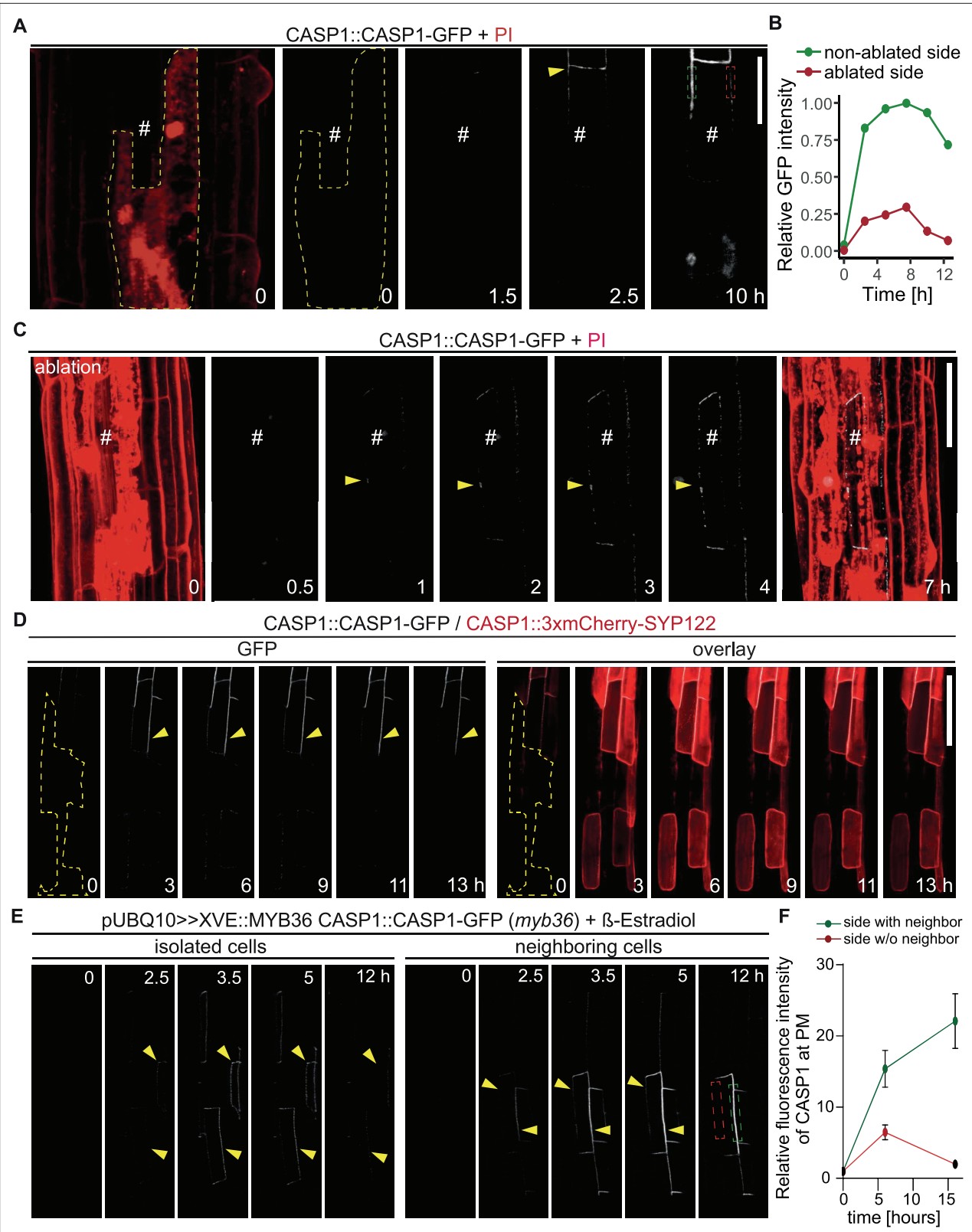

**Figure 3.** Endodermal neighbors are required for proper CSD establishment. (**A**) Endodermal cell ablation leads to instability of the CSD in membranes facing dead cells. Endodermal cells were ablated 3–5 cells prior to onset of CASP1-GFP (white) expression and followed over indicated time spans. First image depicts overview after ablation with cells outlined with PI; # depicts remaining live endodermal cell, dashed yellow line depicts cell outlines of ablated cells, arrow indicates onset of CASP expression, dashed green and red rectangles indicate quantification areas shown in (**B**), scale bar = 20 µm.

*Figure 3 continued on next page*

*Figure 3 continued*

(**B**) Quantification of GFP fluorescence over time in membranes facing live (green) and dead (red) adjacent endodermal cells. Intensity depicted in relation to maximum total GFP intensity. (**C**) Ablation of all adjacent endodermal cells prevents fusion of CSDs. Endodermal neighbors were ablated at timepoint 0 and CASP1-GFP expression (white) followed over time in remaining cell (#), PI (red) highlighted cell outlines and confirmed destruction of cells; yellow arrow indicates initial CASP microdomains, scale bar = 50 um. (**D**) CASP instability in membranes facing ablated cells is independent from CASP expression. Cells ablated 3–5 cells prior to onset of CASP expression, cell outline depicted with dashed yellow line, arrow indicates stable CASP formation, scale bar = 100 um. (**E**) CASP fusion occurs only if both neighboring cells reach expression threshold. Endodermal differentiation was induced via 0.5 μM β-estradiol and CASP1-GFP expression followed over the next 12 hr. Yellow arrows indicate CASP deposits in the membrane, dashed green and red rectangles indicate areas for quantification shown in (**F**). (**F**) Quantification of GFP fluorescence over time in membranes reaching local stability threshold (green) and those below the threshold (red).

The online version of this article includes the following figure supplement(s) for figure 3:

**Figure supplement 1.** Cell ablation impacts CS establishment.

## LOTR1 defines a novel pathway controlling CASP domain positioning

While no other knock-out mutant resembles *lotr1*, overstimulation of the SCHENGEN (SGN) pathway by external application of CIF2 peptide, leads to ectopic formation of CASP domains somehow resembling the phenotype of *lotr1*. The CIF2 peptide is perceived by the Leucine-rich repeat receptor kinase SGN3 (also called GSO1), a key receptor controlling a novel apoplastic barrier surveillance pathway (*Doblas et al., 2017*; *Nakayama et al., 2017*). A complete absence of SGN3 activity in knock-out mutants leads to an incomplete fusion of otherwise correctly positioned CASP micro-domains. In order to test, whether *lotr1* ectopic patches are due to an overactive SGN pathway, we generated double knockouts of *lotr1* in combination with *sgn1*, *sgn2* or *sgn3*. Single mutants *sgn1* and *sgn2* showed sporadic disruptions in their central CSD while *sgn3* displayed more numerous holes in comparison, agreeing with previously published data (*Alassimone et al., 2016*; *Doblas et al., 2017*; *Figure 4A*). Double mutants displayed *lotr1*-typical ectopic deposition of CASP1-GFP, as well as discontinuities of the central domain. Although discontinuities in the central CSDs were visible in *lotr1* itself, these occurred mostly at cell junctions, while *sgn*-typical interruptions were present throughout the length of the CSD. Both types of disruptions were found in the respective double mutants. Moreover, no significant difference between the double and single mutants was observed in blockage of apoplastic tracer PI, a test for CS functionality (*Figure 4C*). This clear additivity of phenotypes suggests that LOTR1 and SGN mutants are involved in separate pathways. It certainly excludes a model whereby ectopic patch formation of *lotr1* is due to an overactive SGN pathway. We also tested a possible connection to ESB1. Again, *lotr1 esb1* double mutants displayed an addition of parental phenotypes with an *esb1*-typical string-of-pearl-stage central CSD and *lotr1*-typical ectopic CASP deposits and showed no enhanced apoplastic barrier phenotype, indicating that LOTR1 also acts independently of ESB1 in CS domain formation.

The model of SGN3-CIF1/2 integrity signaling predicts enhanced lignification of endodermal cell corners and an early onset of suberization whenever the diffusion barrier is broken in such a way that CIF peptides cannot be restricted anymore. The previously described enhanced suberin phenotype in *lotr1* is consistent with such a barrier-defect induced activation of the SGN pathway, due to its large corner gaps in the CS domain. The previously reported large gaps in lignification of CSs (*Li et al., 2017*), fit the discontinuous CSDs in this mutant. Presence of KMnO$_4$-stained lignin throughout the endodermal cell corners found in some samples during our ultra-structural analysis (*Figure 1—figure supplement 1E*) resembled SGN3-induced ectopic lignification. To unambiguously demonstrate ectopic lignin deposition outside the CASP-labeled domain, we employed a novel clearing protocol that enables co-observation of fluorescent markers, for example CASP1-GFP, and lignin-specific stains, such as Basic Fuchsin (*Ursache et al., 2018*; *Figure 4—figure supplement 1A*). As described, in addition to the central CS, lignin accumulates at ectopic CASP1-deposits in *lotr1* (*Figure 4B*). However, cell corner lignification not associated with CASP1 domains was observed at sites of gaps in the CSD, but not at places where the CSD was continuous. This nicely supports the recently proposed model in which SGN3-stimulation can induce local lignification with subcellular precision (*Fujita et al., 2020*). Consistently, enhanced lignin deposition in cell corners was not present in the *lotr1 sgn3* double mutant, confirming that this ectopic lignin is caused by activation of the barrier surveillance mechanism. *lotr1*-typical patches, by contrast, were still lignified in the double mutant, supporting the concept of two separate pathways controlling lignin deposition in the endodermis: A developmental

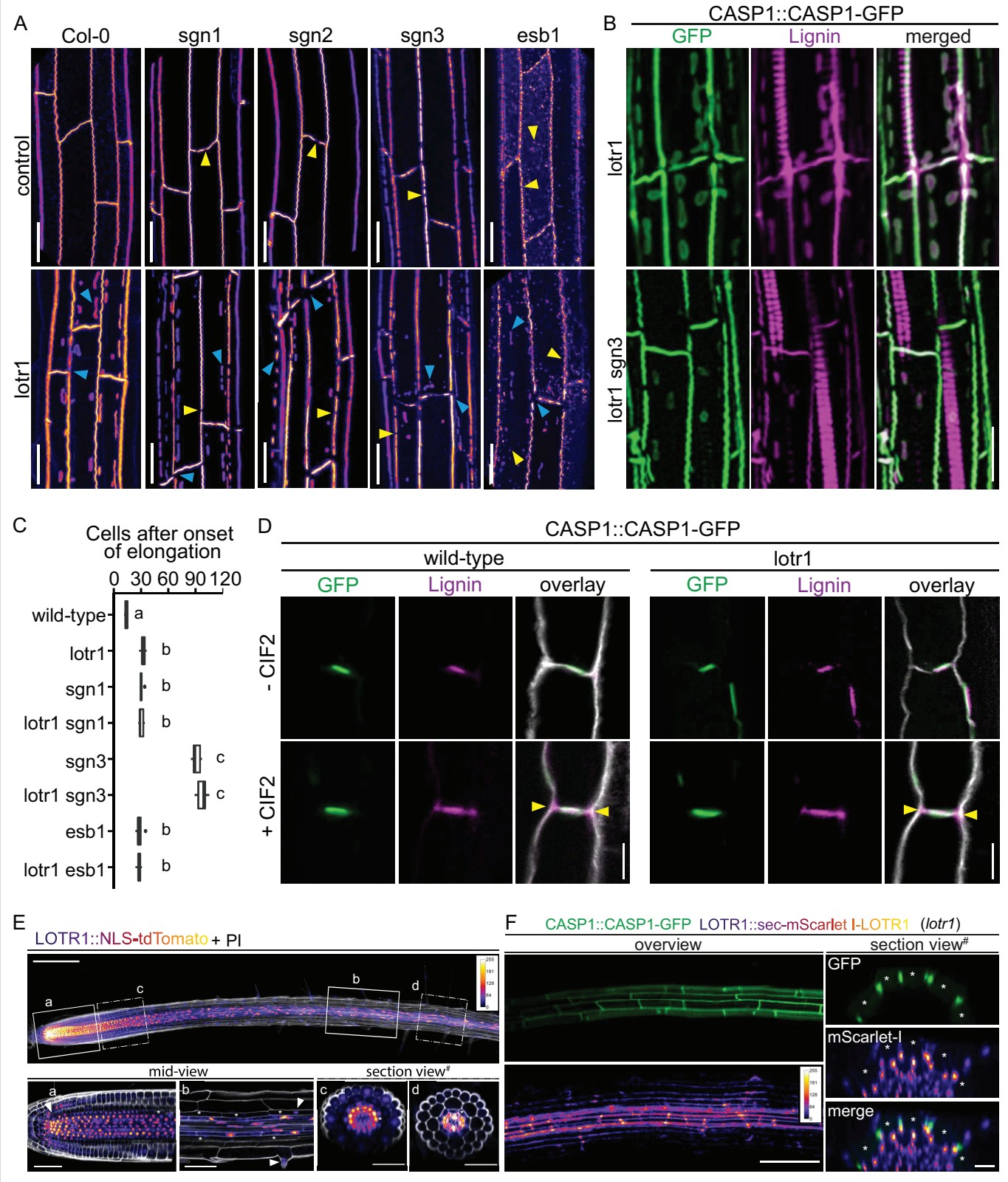

**Figure 4.** LOTR1 restricts CSD establishment through independent pathway. (**A**) CASP1-GFP fluorescence of known CS mutants in wildtype and *lotr1* background. Yellow arrows indicate parental CS mutant phenotype while blue arrows highlight *lotr1* specific phenotypes; scale bar = 20 µm. (**B**) Enhanced lignification in *lotr1* is SCHENGEN-dependent. Ectopic lignin (magenta) accumulates in *lotr1* at disruptions of the CASP1-GFP (green) marked CSD. Lignification of the central domain and ectopic patches is unaffected by disruption of *SGN3*, scale bar = 20 µm. (**C**) Apoplastic barrier

*Figure 4 continued on next page*

*Figure 4 continued*

phenotypes of *lotr1* double mutants. Formation of apoplastic barrier was determined by penetrance of apoplastic tracer PI to the stele; n = 10, significance determined by one-way ANOVA and separate groups identified with Tukey-Kramer test. (**D**) SCHENGEN-pathway is unaffected in *lotr1*. Wildtype and *lotr1* seedlings were treated for 24 hr with 100 µM synthetic CIF2 peptide and lignin accumulation (magenta) outside the regular CSD domain (green) analysed. Cell walls were stained with Calcofluor White, yellow arrows indicate ectopic lignin accumulation, scale bar = 5 µm. (**E**) LOTR1 expression indicates additional roles. Promoter activity highlighted by tdTomato fluorescence in 5-day-old seedlings; cells outlined by Propidium iodide (white); mid-view regions depicted below highlighted by continuous rectangles (**a,b**) and section views indicated by dash-dotted lines (**c,d**), inlay shows color profile of *LOTR1* expression intensity. a Single-plane section through root meristem region, arrow indicates cortex-daughter cell. b Single-plane section after cell elongation, arrows indicate expression in endodermis (top arrow) and epidermis (bottom arrow). c Maximum-intensity projection of section view of z-stack through meristem region indicated in top picture. d Maximum-intensity projection of section view of z-stack through mature region indicated in top picture; * denotes endodermal cell lineage, scale bars: *top*: 100 µm; *a,b*: 20 µm; *c,d*: 50 µm. (**F**) Complementation of *lotr1* phenotype. N-terminal fusion construct expressed under native *LOTR1* promoter complements ectopic CSD phenotype. Scale bars: *left* 100 µm, *section views* 20 µm.

The online version of this article includes the following figure supplement(s) for figure 4:

**Figure supplement 1.** Investigation of putative post-translational processing in LOTR1.

lignification pathway building up CS-type lignin and the SGN3-mediated ectopic lignin deposition that compensates in case of barrier defects. Exogenous application of CIF2 peptide further demonstrated that the SGN3-pathway is functional in *lotr1* (*Figure 4D*). In the presence of CIF2, both genotypes formed ectopic lignin at cell corners.

## LOTR1 is broadly expressed in the root

Due to a lack of available expression data in public databases (*LOTR1* was not included on the ATH1 microarrays), it was crucial to generate a reporter line. We therefore placed a nuclear-localized tdTomato under the control of a 3 kb LOTR1 promoter fragment. We observed strong promoter activity in the root meristem in nearly all tissue layers and cells, including QC and columella, with the highest expression found in the stele (*Figure 4E*). Very low expression was found in the epidermis and ground cell lineage, with promoter activity visible in cortex/endodermal initials and their endodermis daughter cells, but not in the cortex daughter. In older meristematic cells, *LOTR1* expression further reduced in epidermis and endodermis, whereas stele cells continued to show strong promoter activity. This pattern continued past the elongation zone, where stele expression was still strong with some residual signals found in epidermal and endodermal cells. Inside the stele itself, *LOTR1* was specifically absent in xylem precursor and xylem-adjacent procambium cells, whereas pericycle, phloem precursor and phloem-adjacent procambium cells displayed strong expression (*Figure 4E*). Recently published single-cell RNA sequencing results showed strong expression of LOTR1 in clusters corresponding to phloem, stele, xylem, and root cap cells, confirming our promoter activity analysis (*Ryu et al., 2019*; *Denyer et al., 2019*). Despite this broad expression pattern, *lotr1* mutants displayed only phenotypes related to endodermal differentiation, that is ectopic CASP deposition and disrupted CSDs causing compensatory lignification and suberin deposition, which themselves are likely responsible for the previously observed delay in lateral root development and low-calcium-sensitivity phenotype (*Li et al., 2017*). Consequently, if the expression pattern points to a much broader function for LOTR1 outside of endodermal differentiation, these additional functions are probably redundantly fulfilled by some of the many homologs of LOTR1 (*Figure 4—figure supplement 1E*). *LOTR1 is part of a large plant family of putative, extracellular proteases.*

*LOTR1* codes for a putative cell wall protein with a predicted signal peptide, a putative auto-inhibitory domain (Neprosin-AP), and a Neprosin domain, recently identified as a novel proline protease domain in an enzyme found in the digestive fluids of pitcher plants (*Rey et al., 2016*; *Schräder et al., 2017*). The numerous EMS Alleles of LOTR1 all appear to be loss-of-function alleles of equal strength and consist of STOP mutations all along the ORF, as well as several point mutations in the Neprosin domain (*Figure 4—figure supplement 1B*). We found 42 other proteins in the *Arabidopsis* proteome with a domain structure like LOTR1, that is having both Neprosin-AP and Neprosin domains, although subsequent phylogenetic analysis showed considerable variation among the family members. Average identity towards LOTR1 among this 'HOMOLOGS-OF-LOTR1' (HOLO) family was only 38% ( ± 12) with the three closest homologs standing out at ~70% (*Supplementary file 1*). Phylogenetic clustering identified five subgroups (A:E) with instances of gene-duplication events visible

especially in group C, where eight homologs were found in consecutive loci on chromosome 4, highlighting the variable nature of this family (*Figure 4—figure supplement 1E*). Nevertheless, putative orthologs of LOTR1 can be identified in both mono- and eudicot species, indicating that, although some family members seem to evolve rapidly, LOTR1 has a deep conservation of function, expected for a regulator of CS formation.

While 34 of the putative homologs showed predicted signal peptides for apoplastic localization, we also found five proteins in the family predicted to have a single N-terminal transmembrane domain with their active domain outside in the apoplast (*Figure 4—figure supplement 1E*, *Supplementary file 1*). Thus, the HOLO family are interesting candidates for proteolytically processing membrane or cell wall proteins relevant for domain establishment. Single-cell RNAseq expression profiles of *LOTR1's* four closest homologs partially overlapped in cell clusters for QC/meristem (At1g23340, At1g70550, At5g56530), root cap (At1g10750, At1g70550), phloem (At1g70550, At5g56530), and xylem (At1g10750, At23340, At1g70550, At56530) (*Ryu et al., 2019*), explaining the lack of any meristematic phenotypes in *lotr1*. However, while eight separate mutants with this unique patchy phenotype, that were identified in the *LORD-OF-THE-RINGS* screen, all turned out to be alleles of *LOTR1*, none of its homologs were found, indicating they likely have separate roles.

Mature and active Neprosin has been reported to consist only of its functional domain (PF03080) (*Rey et al., 2016*), presumably undergoing processing during or after secretion. However, only full-length protein was found in functional, full-length complementation lines (*Figure 4—figure supplement 1*), suggesting that LOTR1 might retain its putative auto-inhibitory Neprosin-AP domain. Expression of artificially truncated versions of LOTR1 consisting of only its Neprosin domain or only its putatively auto-inhibitory Neprosin-AP domain resulted in extensive intracellular labeling and absence of complementation (*Figure 4—figure supplement 1D*), supporting that LOTR1 does not undergo post-processing for activation.

As many of the Neprosin-containing proteins are annotated as putative carboxyterminal proteases, we investigated potential homologies towards such enzymes. Database searches identified *E. coli* protein mepS, a cell wall protease required for cell wall expansion, as a potential homolog. Although displaying only 20% identity to LOTR1's Neprosin domain, structural comparison revealed that three crucial amino acids in the catalytic center of mepS, one cysteine and two histidines, are present in LOTR1, its three closest homologs, as well as Neprosin itself (*Figure 5—figure supplement 1B,C*). Expression of a full-length LOTR1 version with putative active-site residues C255 and C260 mutated to alanine showed apoplastic localization similar to unmutated LOTR1, but was unable to complement the *lotr1* phenotype (*Figure 4—figure supplement 1D*). This suggests an important functional role for these amino acids in LOTR1 and supports a protease function for this enzyme class in *Arabidopsis*.

## Tissue-specific complementation indicates non-cell autonomous LOTR1 action

The strong expression pattern of LOTR1 suggests a function in the inner stele tissues and a non-autonomous action of LOTR1 on endodermal differentiation. With a potential protease function, an action on the equally stele-expressed CIF peptides was conceivable, but inconsistent with our genetic analysis. In order to establish the site of LOTR1 action, we generated LOTR1 fusion proteins and were able to complement the loss-of-function phenotype with LOTR1 N-terminal (behind the signal peptide), but not C-terminal fusion proteins, indicating that a free C-terminus is important for its functionality (*Figure 4F*). Interestingly, the *lotr1-8* allele, where a late STOP-codon truncated the protein by just 16 aa leads to a knock-out phenotype, further suggesting an important role of the c-terminal extremity. In agreement with the promoter activity, fluorescence of the fusion construct was mainly visible in the apoplastic space between stele cells, with outer tissues showing very low levels of accumulation (*Figure 4F*).

To determine if the very low expression observed in the endodermis is nonetheless responsible for LOTR1 function, we generated lines driving LOTR1 expression from stele-, endodermal-, or cortex-specific promoters in the *lotr1* mutant background. We found continuous CSDs and lack of patches only when expressed from the stele specific SHORTROOT (SHR) and CIF2 promoters, but not from endodermal SCR, CASP1, and ELTP, nor cortex specific C1 promoters (*Figure 5A*). Covering early, intermediate, and late endodermal development, the lack of complementation by endodermal promoters demonstrated that LOTR1 was not acting in the endodermis to fulfil its role in CSD

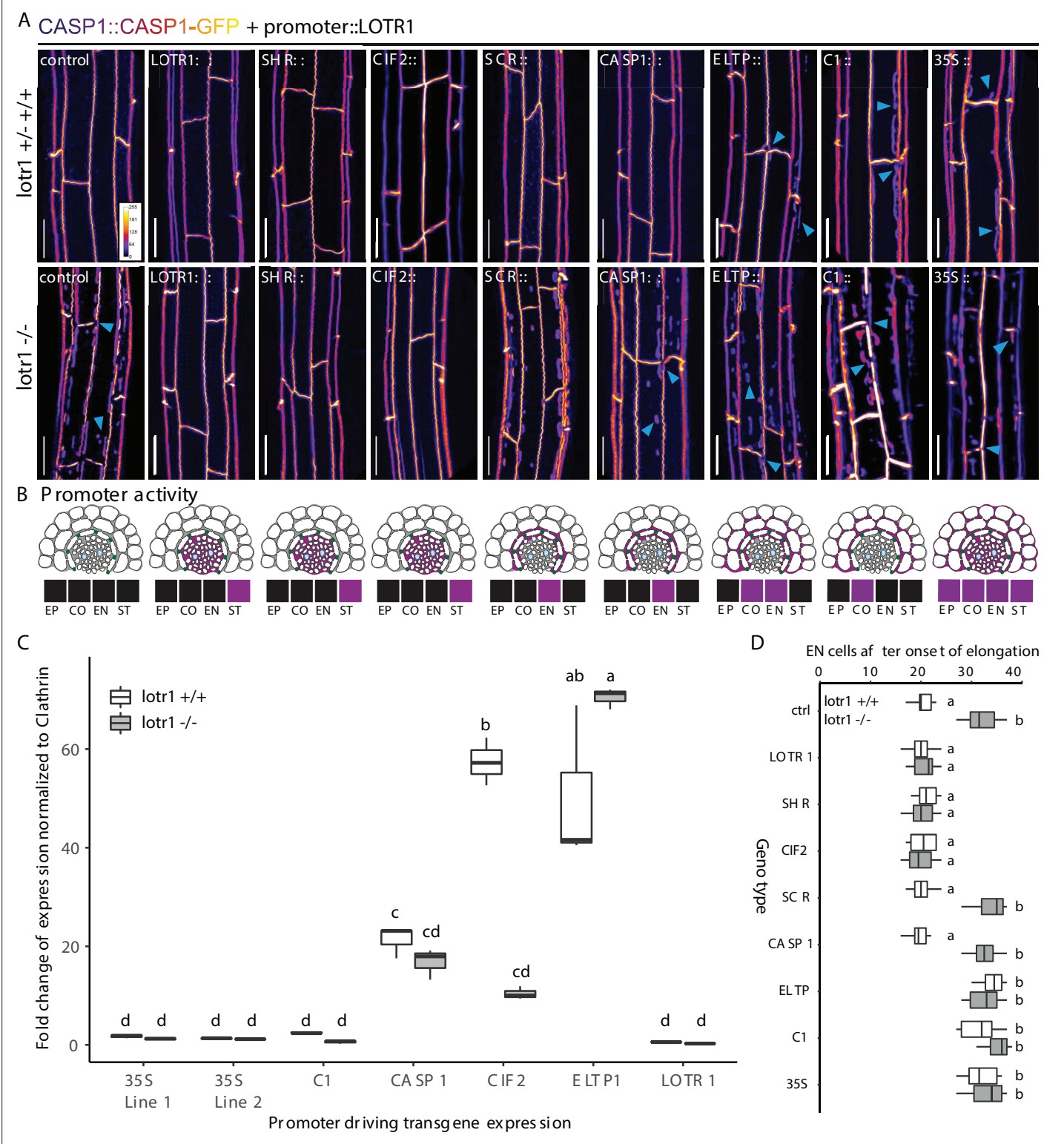

**Figure 5.** Stele-specific LOTR1 restricts CSD establishment. (**A**) Mis-expression of LOTR1 in cortical cells causes dominant loss-of-function phenotypes. CASP1-GFP phenotypes of seedlings expressing LOTR1-complementation construct (used in *Figure 4F*) under indicated promoters. Presented pictures represent typical phenotypes observed in more than five independent lines; blue arrows indicate *lotr1*-like phenotypes; EP = epidermis, CO = cortex, EN = endodermis, ST = stele; scale bars = 20 μm. (**B**) Schematics of promoter activity to highlight cell-type specific expressions are based on publicly available microarray expression data obtained from the BAR eFP browser (*Winter et al., 2007*), as well as diverse publications that provide detailed characterisations of these promoters Promoters are as follows: *SHR, SHORT ROOT; CIF2, CASPARIAN STRIP INTEGRITY FACTOR 2;SCR,*

*Figure 5 continued on next page*

*Figure 5 continued*

SCARECROW; CASP1, CASPARIAN STRIP DOMAIN PROTEIN 1; ELTP, LTPG15, LIPID TRANSFER PROTEIN G 15; C1, CORTEX 1. Expression indication for LOTR1 was modeled according to the patterns shown in *Figure 4E/F*. (**C**) Transgene expression comparison of lines presented in (**A**) by qRT-PCR. The data presented here are representative of three biological replicates (n = 3) analysed. NOTE: Transgene expression levels do not correlate with occurrence of dominant phenotypes observed in **A**. Statistical difference was determined using ANOVA to search for significant interactions between expression level, transgene presence and genetic background. Significant interactions were grouped by a post-hoc Tukey-Kramer test and depicted with letters. (**D**) Apoplastic barrier phenotypes of tissue specific complementation lines in wildtype (white) and *lotr1* (grey) background. Cells after onset of elongation counted until penetrance of PI was blocked. n = 10, significance determined via one-way ANOVA and independent groups identified via Tukey-Kramer post-hoc test.

The online version of this article includes the following figure supplement(s) for figure 5:

**Figure supplement 1.** Neprosin domain analysis in the HOLO family.

formation. Peculiarly, expression of the same construct under the ubiquitous 35 S CaMV promoter did also not seem to complement the underlying *lotr1* phenotype. Several scenarios could explain this finding: LOTR1 function could require exclusive expression in the stele, although its native promoter showed some weak outer-tissue activity. Alternatively, over- or misexpression of the construct might lead to the same patchy *lotr1* phenotype, but through dominant interference. We therefore analysed the expression levels of the via qRT-PCR. We observed high transgene expression levels in lines expressing *LOTR1* from endodermal/cortical promoter ELTP and stele-specific CIF2 (*Figure 5C*). Expression strength of *LOTR1* expressed from 35 S and C1 promoters were comparable to the native promoter. Thus, overexpression does not explain the seeming absence of complementation in the 35 S lines, leaving mis-expression in outer tissues as the probable cause for the observed phenotype.

We therefore tested whether ubiquitous expression via 35 S would lead to a dominant-interference phenotype. We outcrossed all generated lines to wildtype CASP1-GFP plants to test for genetic dominance of the transformed constructs. Previously complementing constructs LOTR1::LOTR1, SHR::LOTR1, and CIF2::LOTR1 continued to display a wild-type CSDs and no occurrence of *lotr1* typical patches in the F1 generation (*Figure 5A*). Expectedly, wild-type phenotypes were also observed for the non-complementing, endodermis-specific SCR and CASP1 driven lines. Yet, remarkably, not only expression driven by 35 S, but also from cortex-specific C1 and endodermal/cortex-expressing ELTP promoters caused ectopic CASP1-GFP patches like those observed in uncomplemented *lotr1* in the F1 progeny. We confirmed this dominant phenotype in the following segregating F2 generations, where all plants harboring ELTP, C1, or 35 S constructs showed patch formation. Only 7/37 (ELTP), 9/42 (C1), and 8/45 (35S) plants displayed wildtype like continuous CSDs and no patches in the respective F2 populations. Since the dominant phenotype was not easily distinguishable from the normal *lotr1* loss-of-function phenotype, our observed ratios (18.9%, 21.4%, 17.7 %) matched the expected 3/16 (18.75%) Mendelian segregation ratio for a wildtype phenotype (non *lotr1*-homozygous, plus absence of dominant transgene). Apoplastic barrier functionality tests in these lines confirmed our observations (*Figure 5D*). Finally, we ensured by qRT-PCR that expression from the native locus was largely unchanged in all transgenic lines excluding transgene-induced silencing as a possible explanation for the observed, dominant-negative effects (*Figure 5—figure supplement 1A*).

## Discussion

Establishment of a functional apoplastic barrier relies on the correctly localized deposition of lignin at the median position between endodermal cells (*Naseer et al., 2012*; *Geldner, 2013*). Although many components involved in this process have been identified in the last decade, we still poorly understand the mechanism by which these cells achieve the precise formation of Casparian Strips in such a way that a single, thin, continuous ring of lignin is formed in between endodermal cells. In our current work, we provide new insights into this mechanism by demonstrating strict coordination of CASP deposition in membranes of adjacent endodermal cells. Independent of any known CS mutant, this coordination pathway appears to allow a local initial pairing of forming CASP micro-domains, across the cell wall of neighboring endodermal cells, which promotes their stabilization and eventual fusion. This results in the necessity for having a functional, differentiating endodermal cell neighbors in order to form a continuous CS. Without a neighboring cell in the same developmental state, wild-type endodermal cells abort CS development after the initial formation of CASP

microdomains, which eventually disappear. In our current understanding, CASPs are initially secreted in a non-localized fashion (*Roppolo et al., 2011*), yet rapidly form aligned CASP micro domains. This is initiated by a specific exocyst complex, defined by subunit EXO70A1, whose accumulation at the site of the future CS precedes CASP accumulation and is necessary for their targeted secretion to this domain (*Kalmbach et al., 2017*). Accordingly, *exo70a1* mutants display a high number of non-localized and non-functional, unstable CASP microdomains. What mechanism could account for the selective stabilization of CASPs in domains facing intact neighbors of the same differentiation state, but not toward dead, undifferentiated cells or cells of a different cellular identity? One explanation would postulate the existence of a stabilizing short-range signal, produced by CASPs domain that can only act in trans, not in cis. However, such an exclusive in trans action is difficult to conceive with small, released molecules able to diffuse across the cell wall space. It would be much easier to conceive with filaments, reaching across the cell walls between domains, but there is no evidence for the existence of such filament-like structures. Alternatively, a cumulative threshold for a short-range, stabilizing signal produced by the microdomains can be postulated, that could only be reached when both cells produce it. However, such a model would require a very fine-tuned thresholding. Although a promising candidate for a factor involved in this coordination process, our detailed analysis of the *lotr1* mutant demonstrates that it does not affect the coordination of the properly localized micro-domains between cells, nor does it allow for their formation without neighbors. Nevertheless, absence of LOTR1 activity clearly allows for formation of non-paired, stable, and fully functional microdomains, oriented towards non-endodermal neighbors. Therefore, it is most likely responsible for inhibiting sporadic micro-domain formation throughout the endodermal membrane when CASPs are starting to focalize from their ubiquitous distribution in the plasma membrane into aligned micro-domains. The peculiar formation of Casparian half-strips in *lotr1*, further indicates that lignin deposition into the CS cell wall is separately performed by both contributing cells, which would make a tight coordination of neighboring CASP domains all the more critical for achieving a fully sealed cell wall space. Restricted mostly to the stele in mature regions of the root, we provide evidence that this putative cell wall protease might act on a target originating outside of its own expression domain. *lotr1* could only be complemented using stele-specific promoters and neither endodermal, cortex nor ubiquitous expression leads to complementation. Importantly, we discovered that wildtype plants displayed a dominant *lotr1*-like phenotype with ectopic patches, disrupted CSDs, and a corresponding delay in PI block when LOTR1 was mis-expressed in the cortex. How could ectopic expression of a putative cell wall protease in the cortex interfere with the action of the same protease in its wild-type expression domain in the stele? The most parsimonious, albeit speculative, explanation we can conceive is that a LOTR1 substrate is produced in the outer cortical tissues and has to diffuse into the stele for activation by stele-expressed LOTR1 (*Figure 6*). Expression of LOTR1 in the cortex would lead to a premature activation of its substrate in the cortex and not allow it to reach the stele. It would thus deplete the endogenous LOTR1 of its substrate and cause the observed dominant loss-of-function phenotype. This explanation thus draws a 'reverse' SGN3/CIF pathway model where a cortex-specific substrate passes across the endodermal cell layer to be activated by a stele-specific enzyme complex, destabilizing CASPs and preventing domain formation at ectopic positions.

## Materials and methods
### Plant material and growth conditions
All experiments were performed in the *Arabidopsis thaliana* Columbia-0 ecotype. T-DNA insertion lines obtained from NASC: *lotr1-10* (SALK_051707) (*Li et al., 2017*), *sgn3-3* (SALK_043282) (*Pfister et al., 2014*), *sgn1-2* (SALK_055095 C) (*Alassimone et al., 2016*), *sgn2-2* (SALK_009847) (*Doblas et al., 2017*); *myb36-2* (GK- 543B11) (*Kamiya et al., 2015*), *esb1-1* was kindly shared by Prof. David Salt's group (*Hosmani et al., 2013*), and the *lotr1-1* allele was identified in the *LOTR*-screen (*Li et al., 2017*). Inducible pUBQ10>> XVE::MYB36 were obtained from the TRANSPLANTA collection (N2102512 and N2102513) and crossed to CASP1::CASP1-GFP. If not otherwise stated, plants were grown as follows: Seeds were surface sterilized and sown on half-strength Murashige-Skoog medium (Duchefa) solidified with 0.8% plant agar (Duchefa). After stratification for 2 days at 4 °C, seeds were germinated in growth chambers with long-day conditions (16 h light/8 h dark) and temperature cycling between 22 °C/19 °C (day/night). 5 days after germination, seedlings were directly analysed or cleared and

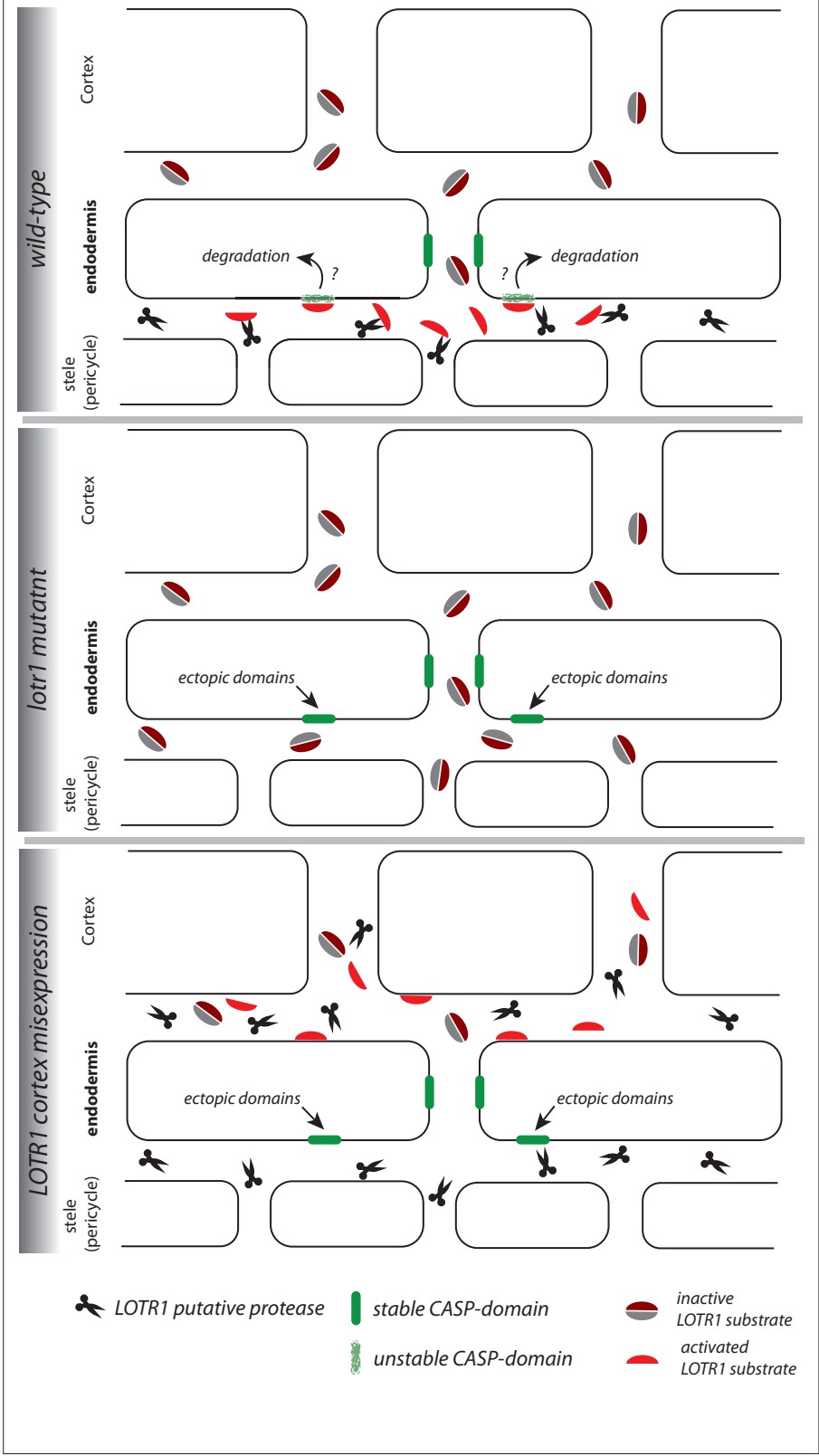

**Figure 6.** Speculative model of LOTR1 function. The following model for LOTR1 function is simply one scenario that can plausibly integrate the data on LOTR1's predicted activity, expression, subcellular localization, as well as phenotypes by loss-of-function and dominant-interference by misexpression. **Wild-type:** LOTR1 is a predicted protease (*scissors*), experimentally determined to be expressed in the stele and to localize in the cell wall. We

*Figure 6 continued on next page*

*Figure 6 continued*

speculate that LOTR1 cleaves a cortex-derived substrate (*grey, dark-red coffee bean, inactive*), activating it in the stele (*red half-bean, active*). This substrate then inhibits ectopic CASP-domain formation (*green*) at the stele-facing endodermal surface by unknown means. *lotr1* mutant: Absence of LOTR1 would not allow activation of the ectopic CASP-domain inhibitor in the endodermal, stele-facing apoplast, leading to the observed formation of ectopic, stable CASP-domain predominantly at the stele-facing side of the endodermal surface. **LOTR1 cortex mis-expression:** This model explains the observation that cortical mis-expression of LOTR1 dominantly interferes with wild-type LOTR1 action, if it would precociously cleave and activate the LOTR1 substrate, not allowing it to reach the stele to be activated by wild-type LOTR1. This would lead to the observed, similar phenotype than the *lotr1* knock-out.

stained with ClearSee as described (*Ursache et al., 2018*). For long-term imaging, agar blocks with seedlings were cut and transferred into microscope chamber slides (*Marhavý and Benková, 2015*; *Marhavý et al., 2016*) and equal volume of double-concentrated β-estradiol added before imaging.

## Plasmid construction

A 3 kb promoter fragment of *LOTR1* was generated by 5'-aacaggtctcgacctctttctatctgtttgtacctaagt-3' and 5'-aacaggtctcatgttaactagagaatgtacggcttgttt-3' and cloned via Eco31I-restriction sites into Green-Gate entry module vector pGGA000 (*Lampropoulos et al., 2013*). Genomic fragment of *LOTR1* coding sequence was amplified with 5'-aacaggtctcaggctcgatgTCAGCTATTCATCTTAAAAACCAAACTTCA and 5'-aacaggtctcaCTGAAGGACACCTTGGGTTCCG-3' and cloned into pGGC000. mScarlet-I was amplified from pmScarlet-I_C1 (*Bindels et al., 2017*) obtained from Addgene and secretion peptide was added by tandem PCR amplification with 5'-gctctttccctctatctcctgcccaatccagccactagtATGGTGAGCAAGGGCGAG-3' (FW1) and 5'-aacaGGTCTCaAACAatgaaagccttcacactcgctctcttcttagctctttccctctatctcctg-3' (FW2) and 5'-aacaGGTCTCaAGCCCTTGTACAGCTCGTCCATGC-3' (RV) and cloned via Eco31I sites into pGGB000. Final destination clones were obtained by subsequent GoldenGate reaction with pGreenII based pGGZ003 and transformed into *Arabidopsis* using floral dip.

## Confocal imaging

Imaging was performed on Leica SP8 or Zeiss LSM880 microscopes, excitation and emission as follows (excitation, detection): Calcofluor White (405 nm, 425–475 nm), GFP (488 nm, 500–550 nm), mCherry/tdTomato/mScarlet-I (563 nm, 580–650 nm), PI (563 nm, 580–650 nm), basic Fuchsin (563 nm, 580–650 nm).

## PI assay

PI was performed as described in *Lee et al., 2013* with following changes: Seedlings were incubated in 10 µg/ml PI dissolved in water for 5 min and imaged immediately afterwards. Number of endodermal cells determined after onset of elongation (defined as first endodermal cell with length to width ratio of at least 2, i.e. at least twice as long than wide).

## Transmission electron microscopy (TEM)

5-day-old *Arabidopsis* seedlings were fixed in glutaraldehyde solution (EMS, Hatfield, PA) 2.5% in 100 mM phosphate buffer (pH 7.4) for 1 hr at room temperature. Then, they were post-fixed in osmium tetroxide 1% (EMS) with 1.5% of potassium ferrocyanide (Sigma, St. Louis, MO) in phosphate buffer for 1 hr at room temperature. Following that, the plants were rinsed twice in distilled water and dehydrated in ethanol solution (Sigma) at gradient concentrations (30% 40 min; 50% 40 min; 70% 40 min); two times (100% 1 hr). This was followed by infiltration in Spurr resin (EMS) at gradient concentrations (Spurr 33% in ethanol, 4 hr); Spurr 66% in ethanol, 4 hr; Spurr two times (100% 8 hr) and finally polymerized for 48 hr at 60°C in an oven. Ultrathin sections 50-nm-thick were cut transversally at 2 mm ±0.2 mm from the root tip, on a Leica Ultracut (Leica Mikrosysteme GmbH, Vienna, Austria) and two consecutive sections were picked up on a nickel slot grid 2 × 1 mm (EMS) coated with a polystyrene film (Sigma).

## Lignin staining with permanganate potassium (KMnO$_4$) using TEM

Visualization of lignin deposition around Casparian strip was done using permanganate potassium (KMnO$_4$) staining (*Hepler et al., 1970*). The first section was imaged without any post-staining. Then the sections were post-stained using 1% of KMnO$_4$ in H$_2$O (Sigma, St Louis, MO, US) for 45 min and rinsed several times with H$_2$O. Then the second section of the grid was imaged. Micrographs were taken with a transmission electron microscope FEI CM100 (FEI, Eindhoven, The Netherlands) at an acceleration voltage of 80kV and ×11,000 magnifications (pixel size of 1.851 nm), with a TVIPS TemCamF416 digital camera (TVIPS GmbH, Gauting, Germany) using the software EM-MENU 4.0 (TVIPS GmbH, Gauting, Germany). Panoramic were aligned with the software IMOD (*Kremer et al., 1996*).

## Cell ablation

Endodermal cells were ablated using a MaiTai-SpectraPhysics laser at 800 nm integrated into a Zeiss LSM880 microscope. A ROI outlining the cells to ablate was used and cells were ablated with 2% power and 0.8 µs pixel dwell time. Cell destruction was confirmed using transmitted light and PI staining.

## Western blot

Roots of 5-day-old seedlings grown on ½ MS medium were cut and weighted before shock-freezing in liquid nitrogen. Samples were then homogenized using TissueLyser II (Qiagen). Proteins were then denatured by addition of 3 x (v/w) 1 x NuPAGE LDS sample buffer +50 mM DTT (Invitrogen, NP0007) and heating for 5 min at 70°C. Cell debris was removed by centrifugation at 1'000 *g* for 5 min before separation of proteins on pre-cast 12% Bis-Tris SDS-PAGE gels (iD PAGE, Eurogentec). After electrophoresis, proteins were transferred onto PVDF membrane using Pierce FastBlotter G2 semi-dry blotting. Membranes were then blocked (5% skim milk in TBS) for 1 hr before incubation with anti-RFP antibody (1:1000, Chromotek 6G6) overnight at 4 °C. After washing the membrane three times with 1 x TBS +0.1% Tween (TBS-T), blot incubated for 2 hr at root temperature with anti-mouse-HRP antibody (1:5000). Finally, after washing blot three times with TBS-T, HRP activity was detected using SuperSignal West Femto Kit (Thermo Scientific) and GE ImageQuant LAS 500.

## qRT-PCR analysis

For qPCR quantifications, the plants were grown as described previously. Only root parts (around 20–50 mg) were collected, and total RNA was extracted using ReliaPrep RNA Tissue Miniprep Kit (Promega). Reverse transcription was carried out with PrimeScript RT Master Mix (Takara). All steps were done as indicated in manufacture's protocols. The qPCR reaction was performed on an Applied Biosystems QuantStudio3 thermocycler using a MESA BLUE SYBR Green kit (Eurogentech). All transcripts are normalized to Clathrin adaptor complexes medium subunit family protein (AT4G24550) expression. Primers used:

> m-Scarlet F: GCCCCGTAATGCAGAAGAAG.
> m-Scarlet R: GTGTAGTCCTCGTTGTGGGA
> LOTR1 SP F : GGGCTTATGCTTCTTCTCTCTC
> LOTR1 Exon R: GCGGGTTGTAAATGAGACGG
> Clathrin F: AGCATACACTGCGTGCAAAG
> Clathrin R: TCGCCTGTGTCACATATCTC

## Phylogenetic analysis

Homologs of LOTR1 were identified by searching the *Arabidopsis thaliana* proteome for proteins containing both Neprosin-AP (PF14365) and Neprosin domains (PF03080). Sequence similarity was compared using MUSCLE (*Madeira et al., 2019*). Phyolgenetic analysis was done using the NGPhylogeny.fr online-tool with PhylML+ SMS settings. Final tree was assembled and annotated with iTOL online tool (*Letunic and Bork, 2019*). Conserved residues were identified using pre-aligned sequences and the online-weblogo tool from the University of Berkely (https://weblogo.berkeley.edu/logo.cgi) (*Crooks et al., 2004*).

## Statistical analysis

Statistical analysis was done in R software (*R Development Core Team, 2017*) (https://www.r-project.org/). For multiple group comparisons, one-way ANOVA was performed, and significantly different groups identified with Tukey-Kramer post-hoc test.

## Acknowledgements

We wish to thank the Central Imaging Facility (CIF) and the Electron Microscopy Facility (EMF) of the University of Lausanne for their technical expertise. Valérie Denervaud Tendon for technical assistance. We also thank Hiroko Uchida and Marie Barberon for their graphical work. This work was supported by an ERC Consolidator Grant (GA-N°: 616228—ENDOFUN) and two SNSF grants (CRSII3_136278 and 31,003 A_156261) to NG, and a Federation of European Biochemical Sciences Postdoctoral Long-Term Fellowship to PM.

## Additional information

### Funding

| Funder | Grant reference number | Author |
| --- | --- | --- |
| Schweizerischer Nationalfonds zur Förderung der Wissenschaftlichen Forschung | 31003A_156261 | Niko Geldner |
| Schweizerischer Nationalfonds zur Förderung der Wissenschaftlichen Forschung | 310030B_176399 | Niko Geldner |
| Federation of European Biochemical Societies | | Peter Marhavý |

The funders had no role in study design, data collection and interpretation, or the decision to submit the work for publication.

### Author contributions

Andreas Kolbeck, Conceptualization, Data curation, Formal analysis, Investigation, Methodology, Writing – original draft; Peter Marhavý, Conceptualization, Investigation, Writing – review and editing; Damien De Bellis, Lothar Kalmbach, Investigation, Writing – review and editing; Baohai Li, Investigation, Resources; Takehiro Kamiya, Investigation, Resources, Writing – review and editing; Toru Fujiwara, Project administration, Resources; Niko Geldner, Conceptualization, Funding acquisition, Project administration, Supervision, Writing – review and editing

### Author ORCIDs

Andreas Kolbeck  http://orcid.org/0000-0001-9407-1958
Toru Fujiwara  http://orcid.org/0000-0002-5363-6040
Niko Geldner  http://orcid.org/0000-0002-2300-9644

### Decision letter and Author response

Decision letter https://doi.org/10.7554/eLife.69602.sa1
Author response https://doi.org/10.7554/eLife.69602.sa2

## Additional files

### Supplementary files

- Transparent reporting form
- Source data 1. Source data files.

• Supplementary file 1. Table S1.

### Data availability

All data generated or analysed during this study are included in the manuscript and supporting files.

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
