## [Decision Letter]

**Decision letter after peer review:**

Thank you for submitting your article "CASP microdomain formation requires cross cell wall stabilization of domains and non-cell autonomous action of LOTR1" for consideration by *eLife*. Your article has been reviewed by 3 peer reviewers, one of whom is a member of our Board of Reviewing Editors, and the evaluation has been overseen by Jürgen Kleine-Vehn as the Senior Editor. The following individuals involved in review of your submission have agreed to reveal their identity: Yvon Jallais (Reviewer #2); Marc Somssich (Reviewer #3).

Essential revisions:

Given the importance of Figure 5 to demonstrate non-cell autonomous function, we request that you:

1) Support the dominant loss-of-function effect, evidence of LOTR1 expression in wt and lotr1 in pELTP, pC1 and p35S constructs (presented in panel C and D) using qrt-PCR for example. Alternatively, if Figure 5 C legend is right, and the LOTR1 fusion was used indeed in panel C, please provide evidence of LOTR1 -mScarlet expression domain when expressed under the different promoters. Is the protein/RNA or an upstream regulator non-cell autonomous/mobile? Alternatively, if non of these are possible please moderate the claims made and revise the figure/paper.

2) Verify LOTR1 expression in WT, lotr1, and lotr1/LOTR1xve::LOTR1 (before and after induction) by qRT-PCR.

*Reviewer #1:*

The paper combines excellent cell biology with genetics approaches. The claims of the paper that LOTR1 is a crucial player in a novel mechanism that control CS positioning are well supported by mutant studies, ablation experiments and expression domain analysis. A weakness is the lack of detail on the molecular mechanism that LOTR1 uses to function or signal CS positioning, which defines the non-cell autonomy aspect of the work.

There is excellent work here on the cell biology and progress in dissecting the role of LOTR1 but there is no understanding of the mechanism involving this protease in the control of the casparian strip. Some of the statements need to be reconsidered or moderated to allow for multiple mechanisms/possibilities. The whole non-cell autonomy effect is based on promoter expression analysis but these results are confounded with a dominant phenotype observed with endodermal promoter. Moreover, since LOTR1 does not appear to move from the stele (do the author have any evidence that it does from results presented in Figure 5??), it is assumed that their non-cell autonomous function is through a downstream yet unknown factor. This uncertainty weakens the conclusions. few recommendations below:

1 – Line 419 and beyond: please describe the nature of lotr1 mutation, which domain of the protein are affected? Line 446 indicate that 8 alleles were identified, are these different mutations? Are these mutations all display similar/ different phenotypes that could help dissecting LOTR function?

2 – Line 455: 'artificially truncated versions of LOTR1 resulted in extensive intracellular labelling and absence of complementation', this evidence do not specifically indicate that ' LOTR1 does not undergo post-processing' but that the post-processed version is not the active form.

3 – line 493: and Figure 5, this section is essential to demonstrate non-cell autonomous function for LOTR1 but promoter expression/activity in lotr1 need to be provided to evaluate complementation results. If complementation depends on LOTR1 dosage, as suggested, then expression levels of the LOTR1 when expressed under the different promoters should be determined.

4 – Expression of LOTR1 with the estradiol promoter is also required. It is difficult to distinguish between the need for LOTR1 to be expressed in the stele alone and the dominant loss-of-function effect observed in wildtype and lotr1 with cortical expressing constructs.

5 – The legend of the figure 5 need to explain what the different colours mean? the title should be moderated as, so far, there is no evidence of the presumible cortex-expressed, mobile target and that figure do not demonstrate the there is one. The legend also need to mention the identity of the different construct and how the activity of the used promoters graph (shown between panel C and D) was determined.

6 – The authors indicate that construct in panel A was the same as used in panel C but in A the LOTR1 was with mSCarlet in magenta and CASP1 in green whereas in section C, CASP1 appear in magenta and LOTR1 is not fused to a reporter?

More importantly, if Figure 5 C legend is right, is it possible to see the expression domain of LOTR1-mScarlet when expressed under the different promoters? Is it restricted to promoter activity domains or there is evidence of out of place protein accumulation/expression?

7 – In relation to the same figure, it is confusing that panels are not presented as described in the text, where panel B (inducible construct) is described after panel C.

8 – Regarding LOTR1 function, as protease activity is not demonstrated neither target of this protease or the localisation of this target in cell walls, authors need to understate their claims and consider alternative (if unlikely) mechanisms. Model presented is highly speculative and it should be clear what aspects of the model are substantiated by the data and which aspects are just hypothetical, ideally by presenting alternative models or reference to the figure in the legend. for example there is no requirement for the target in the steele/endodermis being the same one as in the cortex or is there evidence?

9 – In the vein of LOTR1 non-cell autonomous function, have the authors verified if LOTR1 is mobile, either by diffusing in membranes/ apoplast toward the steele/endodermis face or via plasmodesmata?

Another idea to dissect LOTR1 function could be characterising cell wall composition using for example immunoprobes.

Answering these last questions are not required for an excellent publication but are food for thought for future research.

*Reviewer #2:*

Here the authors investigate how cells of the root endodermis precisely position their Casparian Strip domains (CSD) to form a continuous superstructure. They start with a phenotypic analysis of the lotr1 mutant, which shows ectopic CSD formation, notably at the endodermis/pericycle interface. LOTR1 restricts the formation of CSD independently of other known pathways (Shengen pathway, ESB1). A unique feature of the lotr1 mutant is that the ectopic CSDs are functional (i.e. able to guide lignification). However, lignin is polymerized only close to the CSDs (endodermal cells) and not throughout the apoplast (i.e. not all the way to the PM of the pericycle). This is significant in several ways. First, it shows that lignin polymerization is highly local, while it was anticipated that the ROS and monolignols necessary for lignin polymerization were able to diffuse in cell walls, at least to a certain degree. Second, the production of "half" a Casparian strip in the lotr1 mutant shows that each endodermal cell is responsible for lignifying its part of the cell wall, suggesting tight coordination between neighboring endodermal cells. Live imaging of CASP1-GFP shows that this coordination happens very early (at the string-of-pearls stage). Furthermore, local ablation or conditional mosaic expression shows that neighboring endodermal cells are required for forming a continuous and stable Casparian strip (but not to initially form CSD). The nature of the signal between neighboring endodermal cells is unknown and is independent of LOTR1. Finally, the authors analyzed LOTR1 expression and localization. Using complementation assays with a functionally tagged LOTR1 protein (and some mutated forms), they confirm that it is a secreted apoplastic full-length protein with likely protease activity. In addition, LOTR1 is mostly expressed in the stele, not in the endodermis, pointing to a possible non-cell-autonomous action of LOTR1 to restrict CSD positioning. Using cell-type-specific expression of LOTR1, the authors show that (1) LOTR1 is indeed active in the stele, not in the endodermis, and (2) that it has a dominant-negative effect (mimic a lotr1 loss-of-function phenotype in a wild-type background) when expressed in the cortex.

I find this paper outstanding at several levels. First, the findings that it is possible to form a "half" Casparian strip with only half of the cell wall lignified are fascinating. They imply highly restricted lignin polymerization and will urge plant biologists to reconsider preconceived ideas about apoplastic ROS and monolignol diffusion. The corresponding TEM images supporting this finding are breathtaking (although it took me quite some time to understand them). Second, the level of coordination between neighboring endodermal cells is also astonishing, notably the fact that coordination appears to be almost immediate upon CASP1 expression. The cell ablation and mosaic expression of MYB36 are very elegant (and non-trivial) experiments and unambiguously demonstrate the importance of endodermal neighbors in the formation of a continuous Casparian strip. Last but not least, the findings that LOTR1 action is non-cell-autonomous and induces a dominant-negative effect when expressed in the cortex are not only fully supported by the data but also highly intriguing. Indeed, they suggest that LOTR1 in the stele can cleave and activate a mobile signal produced in the cortex. Although still speculative (because the signal remains unknown and LOTR1 protease activity has not been formally demonstrated), this would suggest that there are signals that pass through the endodermal domain in both directions (cortex to stele for the LOTR1-activated signal and stele to endodermis in the case of the CIF/Shengen pathway). Conceptually, the notion that multiple apoplastic signals regulate their very own diffusion by building up and positioning a diffusion barrier is certainly of broad interest and is very likely transposable to many other biological interphases.

*Reviewer #3:*

The formation and correct positioning of the Casparian strip (CS) requires the coordinated activity of several precisely localized proteins, as well as communication between the different cells. There has been quite some progress in recent years on how this precise localization is achieved. The focus of this paper is the establishment of the Casparian strip membrane domains (CSDs), which are the precursors to the CS and define where the CSs are formed. These CSDs are initiated at several locations along the endodermal cells, but are initially instable, unless in the very precise location of the future CS. In this study, the authors investigate the contribution of LOTR1 to this instability of the CSDs outside the CS domain. Endodermal cells of lotr1 mutants form ectopic, stable, lignified CSDs, which furthermore appear to be one-sided (endodermal-sided), if they occur between an endodermal cell and a non-endodermal cell. It therefore appears that LOTR1, a putative protease, is required to restrict CSD formation to the central CS-domain. Curiously, LOTR1 is expressed in the stele, acts in the endodermis, and ectopic expression in the cortex results in dominant-negative effects. The authors come up with a speculative model, in which a mobile protein is expressed in the cortex, diffuses past the endodermis, and is then cleaved/activated by LOTR1 at the endodermis/cortex interface, where this activated protein then acts to destabilize the temporary CSDs at the cell-sides facing the pericycle.

I believe that the main findings from this paper are that (1) CSDs are indeed only stable between two living endodermis cells, indicating that there is communication between these two cells. And (2), that LOTR1 is required to prevent any one-sided CSDs being formed at endodermis to non-endodermis cell-cell interfaces, thereby contributing to the instability of these initial CSDs. For this, LOTR1 acts non-cell autonomously, being expressed only in the stele. The first of these two findings is nicely and elegantly demonstrated using a combination of cell-ablation experiments, and induced endodermal differentiation in some cells, with or without endodermal neighbors. Regarding the function of LOTR1, the authors perform an exhaustive expression analysis with different promoters and genetic backgrounds. The amount of work to establish these two main points is impressive.

It is then a little bit disappointing that the story seems to be a bit open-ended, as the author's final model still is speculative. However, it is also logically supported by the results presented here, and certainly provides a useful working hypothesis for further work.

It would be great to further validate the model of the authors, but at this stage I do not see any obvious experiments that would help to easily solidify or disprove the model with a reasonable amount of work and time required.

The only observation that does not seem to be explained by the results presented here or the model, is the occurrence of ectopic CASP-deposits in lotr1 mutants at the endodermis-cortex side, which the authors also observed, albeit a rare event (lines 136-139).

---

## [Author Response]

Reviewer #1:The paper combines excellent cell biology with genetics approaches. The claims of the paper that LOTR1 is a crucial player in a novel mechanism that control CS positioning are well supported by mutant studies, ablation experiments and expression domain analysis. A weakness is the lack of detail on the molecular mechanism that LOTR1 uses to function or signal CS positioning, which defines the non-cell autonomy aspect of the work.There is excellent work here on the cell biology and progress in dissecting the role of LOTR1 but there is no understanding of the mechanism involving this protease in the control of the casparian strip. Some of the statements need to be reconsidered or moderated to allow for multiple mechanisms/possibilities. The whole non-cell autonomy effect is based on promoter expression analysis but these results are confounded with a dominant phenotype observed with endodermal promoter. Moreover, since LOTR1 does not appear to move from the stele (do the author have any evidence that it does from results presented in Figure 5??), it is assumed that their non-cell autonomous function is through a downstream yet unknown factor. This uncertainty weakens the conclusions. few recommendations below:1 – Line 419 and beyond: please describe the nature of lotr1 mutation, which domain of the protein are affected? Line 446 indicate that 8 alleles were identified, are these different mutations? Are these mutations all display similar/ different phenotypes that could help dissecting LOTR function?

The mutations were originally described in Li et al., Current Biology, 2017. They are indicated in Figure S4B and now mentioned in the text. In addition to a SALK insertion line, the alleles are typical EMS alleles, with a number of STOP mutations along the ORF, as well as several point mutations within the Neprosin domain.

2 – Line 455: 'artificially truncated versions of LOTR1 resulted in extensive intracellular labelling and absence of complementation', this evidence do not specifically indicate that ' LOTR1 does not undergo post-processing' but that the post-processed version is not the active form.

This is correct, we now changed this statement into: “LOTR1 does not undergo post-processing for activation.”

3 – line 493: and Figure 5, this section is essential to demonstrate non-cell autonomous function for LOTR1 but promoter expression/activity in lotr1 need to be provided to evaluate complementation results. If complementation depends on LOTR1 dosage, as suggested, then expression levels of the LOTR1 when expressed under the different promoters should be determined.

See our reply to the editor’s comments, point (1). We have now determined transgene and endogenous expression level for all lines by qPCR and discussed that the data shows that the phenotypes do clearly not correlate with expression strength, but expression pattern.

4 – Expression of LOTR1 with the estradiol promoter is also required. It is difficult to distinguish between the need for LOTR1 to be expressed in the stele alone and the dominant loss-of-function effect observed in wildtype and lotr1 with cortical expressing constructs.

See our reply to the editor, point (2) We have taken out the estradiol promoter data, because of inconsistent results, but we hope that Reviewer 1 agrees that the cell-type specific promoter line data (and the additional expression quantification we provide) does obviate the need for inducible lines.

5 – The legend of the figure 5 need to explain what the different colours mean? the title should be moderated as, so far, there is no evidence of the presumible cortex-expressed, mobile target and that figure do not demonstrate the there is one. The legend also need to mention the identity of the different construct and how the activity of the used promoters graph (shown between panel C and D) was determined.

We agree that the presence of a mobile signal, is no more than a speculation consistent with the presented data. The title is indeed an overstatement and has been changed. The promoter names were added.

6 – The authors indicate that construct in panel A was the same as used in panel C but in A the LOTR1 was with mSCarlet in magenta and CASP1 in green whereas in section C, CASP1 appear in magenta and LOTR1 is not fused to a reporter?More importantly, if Figure 5 C legend is right, is it possible to see the expression domain of LOTR1-mScarlet when expressed under the different promoters?

The legend is correct, the constructs are indeed mScarlet fusions. However, expression from the different promoters does not lead to very strong signals of LOTR1 and localization of LOTR1 in the cell wall makes straightforward imaging for tissue-specific expression rather challenging and difficult to show in single, comparable pictures. Additionally, some promoters, such as *SCR* or *C1*, only lead to detectable LOTR1 signals early on, but nonetheless affect CASP1 localisation at the later stages shown in Figure 5. We therefore decided not to present this data here. We have, however, ascertained expression of all lines by qPCR (Figure 5) and shown that endogenous LOTR1 expression remains unaffected (Figure S5). Moreover, we have only used well-characterised promoters and based our conclusions (-complementation necessitates expression in stele, -endodermis expression does not rescue, -cortical expression had dominant-negative effects) always on results from at least two independent promoters.

Is it restricted to promoter activity domains or there is evidence of out of place protein accumulation/expression?

As indicated in our reply to the editors’ comments, we think that the comparison of our transcriptional, as well as complementing, translational fusions of LOTR1, suggests that there is no RNA or protein movement of LOTR1. Although LOTR1 is broadly expressed under its own promoter, it shows a sharp drop in expression between stele and cortex in differentiated roots. Here, we do not see any indication that the LOTR1 protein would be detected in a more extended domain than the transcriptional reporter. Rather, protein accumulation follows closely the transcriptional profile.

7 – In relation to the same figure, it is confusing that panels are not presented as described in the text, where panel B (inducible construct) is described after panel C.

We are sorry for this. We have now taken out the inducible construct panel and changed the order.

8 – Regarding LOTR1 function, as protease activity is not demonstrated neither target of this protease or the localisation of this target in cell walls, authors need to understate their claims and consider alternative (if unlikely) mechanisms. Model presented is highly speculative and it should be clear what aspects of the model are substantiated by the data and which aspects are just hypothetical, ideally by presenting alternative models or reference to the figure in the legend. for example there is no requirement for the target in the steele/endodermis being the same one as in the cortex or is there evidence?

We agree that our model is still speculative and indicated this in the title. We have now added an additional “disclaimer” in the legend to Figure 6. It reads:

“The following model for LOTR1 function is simply one scenario that can plausibly integrate the data on LOTR1’s predicted activity, expression, subcellular localization, as well as phenotypes by loss-of-function and dominant-interference by misexpression.”

We would like to point out, however, that it is not trivial to come up with alternative, equally plausible scenarios. It is really very hard to explain how LOTR1 activity in cortical cell layers can interfere with the wt LOTR1 activity in the stele. LOTR1 in the cortex must do something that interferes with the ability of LOTR1 in the stele to act. Competition for the same substrate is by far the most simple hypothesis we could come up with.

9 – In the vein of LOTR1 non-cell autonomous function, have the authors verified if LOTR1 is mobile, either by diffusing in membranes/ apoplast toward the steele/endodermis face or via plasmodesmata?Another idea to dissect LOTR1 function could be characterising cell wall composition using for example immunoprobes.

First, our data indicates that LOTR1 is efficiently secreted (Figure 4F) and therefore should not travel through plasmodesmata, which would require cytoplasmic localization. Second, the non-processed form of LOTR1 that appears to accumulate is rather big (Figure S4C) and, from what is known of cell wall size exclusion numbers, should not be able to move through the apoplast as the smaller plant signaling peptides can. This is another reason why we postulate that the agent that acts on CASP1 localisation in the endodermis is a smaller LOTR1 substrate and not LOTR1 itself.

Reviewer #3:[…]It would be great to further validate the model of the authors, but at this stage I do not see any obvious experiments that would help to easily solidify or disprove the model with a reasonable amount of work and time required.

Thank you for this assessment. We continue to work on the project, but we feel that we will need to invest significantly more time and effort until we can further support or disprove the our current model.

The only observation that does not seem to be explained by the results presented here or the model, is the occurrence of ectopic CASP-deposits in lotr1 mutants at the endodermis-cortex side, which the authors also observed, albeit a rare event (lines 136-139).

We agree that our model does not easily account for the ectopic patches at the cortical side, maybe the LOTR1 substrate can diffuse back towards the cortical side of the endodermis and has a minor role in repressing ectopic patches there? But other, currently unknown factors are more important to repress ectopic patch formation at the endodermis-cortex interfaces?